# Integrated and Sustainable Water and Sanitation Systems at Two Rural Sites in South Africa

**Michael John Rudolph * and Evans Muchesa**

Centre for Ecological Intelligence, University of Johannesburg, Johannesburg 2006, South Africa; emuchesa@uj.ac.za
* Correspondence: michaelr@uj.ac.za

**Abstract:** This paper presents a case study of two sites in South Africa: the Phumulani Agri-Village in Mpumalanga, and a school program in the Eastern Cape. The study evaluates the effectiveness of water and sanitation programs in these rural settings. A transdisciplinary literature review was conducted using primary and secondary data sources from the two sites. The data synthesized themes such as integrated water systems, agroecology, community participation, and resource use. The findings provide insights into the water and sanitation status and implementation of interventions. The application of integrated water systems demonstrated the effective use of rainwater harvesting and storage tanks, upgraded pit toilets, innovative stepped platforms, and stormwater retention methods. At Phumulani, old boreholes were repaired, and new ones were drilled, resulting in adequate water yield from five boreholes for agricultural projects and households, positively impacting the community. Plans to improve access and water quality are in place. These interventions underscore the importance of financial investment, human resources capability, infrastructure, and expertise in evaluation. The water systems have contributed to improved access to water, dignity, and health. Future objectives include containerized water treatment plants as sustainable solutions to ensure consistent, clean water for schools, households, agricultural projects, and eco-toilets.

**Keywords:** integrated water system (IWS); sanitation; energy; food; community participation





## 1. Introduction

"Water, an invaluable resource for survival, is fundamental to equitable socio-economic development. Its significance extends beyond a basic human need, intertwining with various rights enshrined in the Bill of Rights, including the right to water, life, dignity, environment, and food" [1]. SDG Goal 6, part of the United Nations' Sustainable Development Goals, focuses on clean water and sanitation. It aims to achieve universal access to safe, affordable drinking water and adequate sanitation and hygiene by 2030 [1]. The goal emphasizes ending open defecation and addressing the needs of vulnerable groups. It also includes targets for improving global water quality by reducing pollution, eliminating dumping, minimizing the release of hazardous materials, and increasing recycling and safe reuse. Additionally, it seeks to enhance water-use efficiency across all sectors, ensure sustainable freshwater supply, and reduce water scarcity. This holistic approach underlines the vital role of water in promoting social equity, economic progress, and environmental sustainability. In Africa, water and sanitation are critical issues, impacting over 300 million people who lack access to clean drinking water and 700 million people who live without adequate sanitation [2]. This lack of access has impeded poverty reduction and economic growth in the region. The situation is exacerbated by factors such as population growth, urbanization, and climate change, leading to both physical and economic water scarcity. Lack of access to safe toilets results in open defecation, contributing to the spread of waterborne diseases. Moreover, the task of collecting water can expose women and girls to the risk of sexual violence [2]. Water supply remains a major challenge in the Eastern Cape. On

5 August 2021, the Minister of Water and Sanitation, Senzo Mchunu, urged stakeholders in the Eastern Cape to ensure that communities have access to water and decent sanitation. He emphasized the need for concrete plans to address water challenges in the province and across South Africa [3]. Community participation plays a crucial role in rural water supply, as it establishes communities as effective decision-making entities via a consultative empowerment process [4]. In a different South African province, the Mpumalanga Provincial Government initiated the "Water for All" program in 2007 to ensure sustainable water provision via multidisciplinary cross-sector involvement. Despite its positive intentions, there was no concrete evidence of any positive outcomes from the project, highlighting the challenges faced in achieving tangible results [5]. Community participation in water, sanitation, and hygiene (WASH) interventions is crucial for the sustainability and impact of water and sanitation projects in South Africa. This active engagement leads to recognizable changes in community health status and behavior, ensuring the longevity of the projects [4,6]. Effective community engagement must be intentional and sustained over an extended period, leveraging existing knowledge systems embedded in local leadership structures, such as schoolteachers, religious leaders, and community leaders [7]. Despite the South African government's recognition of the importance of integrated water resource management (IWRM) and its embedding in the 1998 National Water Act, practical implementation at the grassroots level, especially in schools and community projects, remains minimal [7,8]. The Department of Water and Sanitation (DWS) in South Africa has urged residents to value their sanitation facilities to prevent the spread of diseases in communities. This call coincides with the global commemoration of World Toilet Day, observed annually on 19 November. World Toilet Day aims to raise awareness about the importance of dignified and safely managed sanitation facilities and highlights the need to prevent the spread of illnesses in communities [9,10].

Community water projects in South Africa face various challenges. Financing is a significant issue due to opposition to private participation, cost recovery constraints, high fiscal deficits by the government, and many municipalities lacking the technical and administrative capacity to manage water assets. Governance presents another problem, with the Department of Water and Sanitation grappling with severe institutional and governance issues, compromising its effectiveness. The country's water infrastructure is often inadequate or in need of repair, leading to delays or abandonment of water infrastructure projects. Environmental factors such as droughts, climate change, and the spread of invasive alien plants also hinder these initiatives [11,12].

This paper aims to explore the effectiveness of community water and sanitation programs and their role in fostering community development. These programs are often a cornerstone of public health and quality of life, and understanding how they contribute to the broader socio-economic development of the communities they serve is crucial. The paper assesses the programs' implementation, including their design and outcomes [13]. It also examines whether these programs are designed to maximize their impact and achieve desired outcomes in improving water and sanitation services. Measuring the accessibility and quality of water and sanitation services are key factors in determining the success of these programs [14,15]. The paper further evaluates whether these services are easily affordable and available to all community members and whether they meet the required quality standards.

Community participation is another important aspect explored in this paper. It assesses the level of active community involvement in these programs, which can greatly enhance the effectiveness of these initiatives [16–18]. Finally, the paper examines the sustainability of the solutions provided by these programs. These solutions must be both effective in the short term and sustainable in the long run, ensuring continued benefits for the community for many years [19–22]. By examining these various aspects, this paper comprehensively evaluates community water and sanitation programs and their impact on community development.

A limitation of the study is its focus on water quality and sanitation standards at only two rural sites. Although this focus enabled a detailed description of these case studies, it may not fully represent the diversity and complexity of water quality and sanitation issues in the province or other regions or contexts. Consequently, the recommendations might be restricted to these and other similar rural sites and settings.

The structure of the paper offers a comprehensive description of the two sites. The literature review explores the integrated water system, South African water legislation, community water projects, and water quality classification. It also discusses global water quality and highlights the initiation of various community-based programs. Despite these ongoing efforts, the review underscores the substantial work still to be carried out concerning water systems worldwide.

In the Materials and Methods section, the paper presents a case study approach to examine two sites: the Phumulani Agri-Village in Mpumalanga and a food security school program in the Eastern Cape. The results section presents the study's findings based on the integrated water system. This section describes the work packages and the findings, demonstrating a good understanding and assessment of the water and sanitation systems at the village and schools, including the appropriate implementation of interventions. These findings are further discussed in the subsequent sections.

The paper concludes with key findings and recommendations for future research.

## 2. Study Sites

The Phumulani Agri-Village (PAV) in Belfast, Mpumalanga, South Africa, is a post-mining agroecological village (Figure 1). PAV experiences a subtropical highland climate characterized by mild summers and cool dry winters. The average annual precipitation is 674 mm (27 in), with most rainfall occurring during the summer months. The rapid degradation of the critical Olifants River catchment, caused by coal mining, has been a significant concern near PAV. The coal mining activities in Mpumalanga have led to soil degradation and air and water pollution, severely impacting farming activities and human health. Moreover, the concentration of coal mines in Mpumalanga has resulted in poor living conditions for the communities affected by the mines, with mining companies failing to engage positively with them [12,23–25].

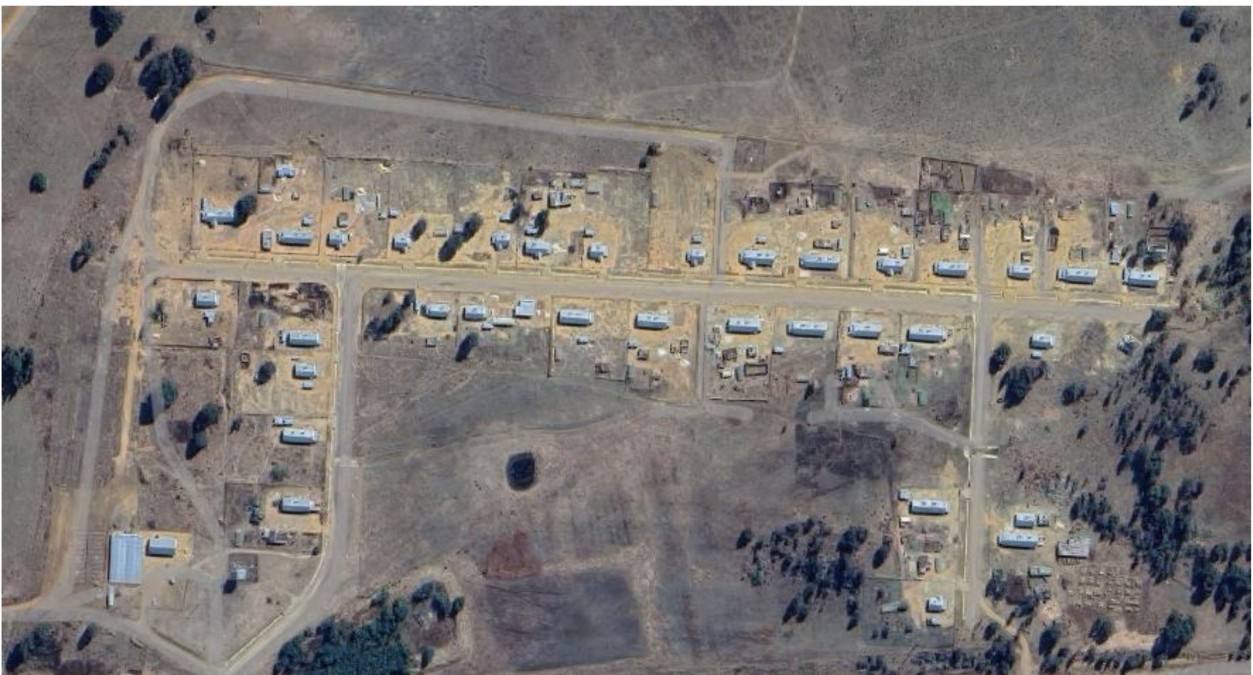

**Figure 1.** Phumlani Agri-Village.

The PAV agroecological project was established to create sustainable jobs and stimulate the local economy by setting up an integrated and circular enterprise. This includes growing and producing a wide range of nutritious organic food, poultry production, seedling propagation, biodigesters, wormery, hemp, and cannabis for local low-income communities. The project also provides training for small-scale and household/community farmers. Its goal is to develop a sustainable post-mining agroecological village, generating jobs and income for the resettled beneficiaries and households, thereby providing sustainable rural livelihoods [26–28].

The Eastern Cape School Program (ECSP) was implemented at Gwebinkundla, Bijolo, and Mqanduli schools, located in the Mqanduli district of the Eastern Cape province (Figure 2). The objective of this program is to establish integrated and sustainable organic food gardens, water, sanitation, and waste management systems. Additionally, it focuses on providing training and capacity building for teachers, learners, and local communities.

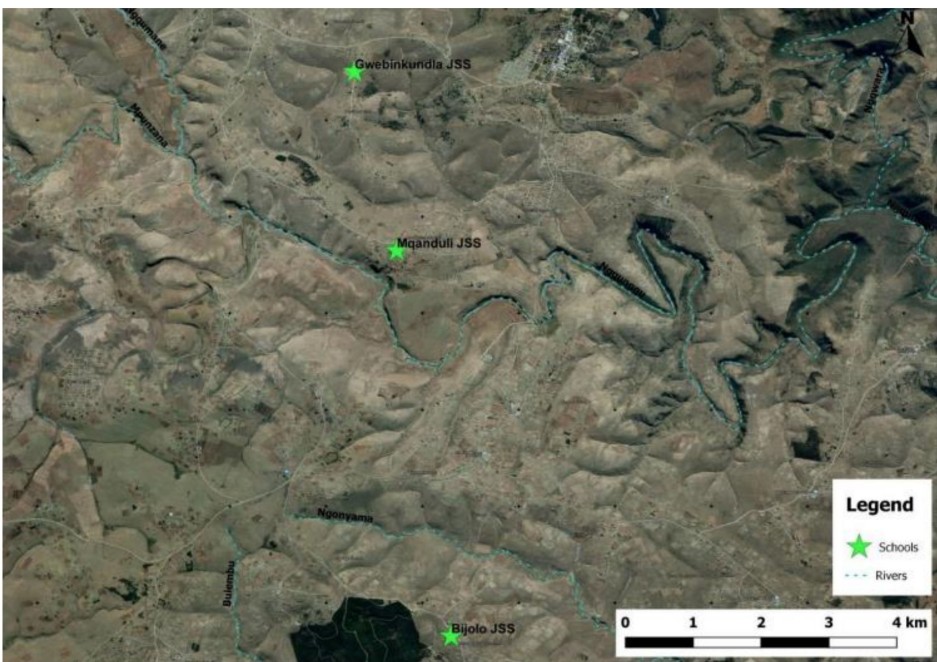

**Figure 2.** Site locality for the EC schools.

Mqanduli town is part of the King Sabata Dalindyebo Local Municipality, which has been severely affected by drought. The persistent drought, coupled with a lack of planning, means that communities in the Eastern Cape, including Mqanduli, struggle daily to access water. Even in areas where the drought is less severe, decades of poor planning and neglect, failure to utilize drought grants, and delays in major infrastructure projects such as dams have contributed to an unprecedented water crisis. Furthermore, the impacts of climate change in the Eastern Cape could lead to rising sea levels, more hot days and heat waves, intensified rains, and overall changes in annual averages. In addition to water scarcity, the region faces challenges related to child nutrition. A recent study conducted in Mqanduli and surrounding areas revealed that their limited access to nutritious food hinders children's healthy growth and development. Unhealthy environments and restricted availability of routine child health services, such as Vitamin A supplementation and deworming, further impede healthy child development [29–33].

## 3. Literature Review

### 3.1. Global Perspective on Water Quality Classification, Legislation Situation, and Community Water Programs

Globally, water quality is categorized into three main aspects: physical, chemical, and biological characteristics. However, the absence of a standardized international system

for this categorization presents a challenge. In the realm of legislation, the Sustainable Development Goals underscore the importance of clean water and sanitation. The United Nations General Assembly has recognized access to safe and clean drinking water and sanitation as a fundamental human right for survival and well-being. However, despite this recognition, there is a pressing need for more concerted efforts to guarantee universal access to safe and clean water. Several community water programs have been launched worldwide to address water and sanitation issues. These initiatives include the Community Water Initiative (CWI), supported by the United Nations Development Program, the Global Water Partnership, and the CDC's global WASH program. These programs aim to mitigate illness by enhancing global access to safe, clean water, adequate sanitation, and improved hygiene. While these initiatives have achieved relative success in their respective domains, much work still needs to be conducted to fully address these challenges fully [34–37].

*3.2. Integrated Water System (IWS)*

An integrated water system (IWS) is an alternative approach that aims to augment the availability of domestic water supply during wet seasons and reduce the water consumption of toilets [14,33]. The three systems include rainwater harvesting system (RWHS), water treatment system (WTS), and eco-toilet system (ETS). The RWHS collects rainwater from rooftops and stores it in tanks for later use. The WTS treats the collected rainwater to make it safe for drinking and other household uses. The eco-toilet system (ETS) is a type of toilet that uses little or no water and instead relies on natural processes such as composting or dehydration to treat human waste. By combining these three systems, an IWS can provide a sustainable water source for households, even in areas with limited water resources [15]. The Water management system (Figure 3) is a comprehensive system designed to monitor and manage water resources that include IWS, WTS, RWHS and ETS systems.

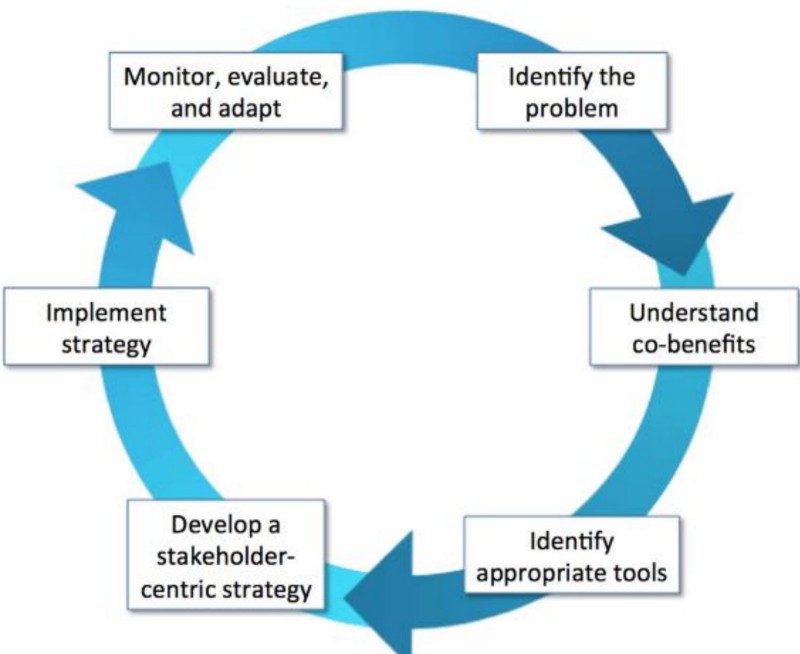

**Figure 3.** Water management system (WMS).

Several water projects are underway or planned in South Africa; some of these are linked to the two projects described [16]:

- Mzimvubu Water Project: Eastern Cape;
- Mqanduli Secondary Bulk Water Scheme;

- The European Partnership called Water4All, which aims to tackle water challenges to address climate change and contribute to achieving the United Nations' Sustainable Development Goals.

In addition to these projects, some of the world's biggest mining companies are working with South Africa's government on a R27 billion water project to supply primary platinum and chrome operations and several hundred thousand people with drinking water [17]. Exxaro, a global coal mining company, has instituted water management and stewardship programs that innovatively manage their water resources. Exxaro's mines, mainly in Mpumalanga, implement their policy via its water management standard covering mining and industrial water use. This standard articulates their integrated water and waste management plan across each mine's lifecycle, including planning, construction, operation, decommissioning, closure, and rehabilitation phases [18].

### 3.3. Legislation on Water in South Africa

The National Water Act (No. 36 of 1998) is the primary legislation governing water management in South Africa. The Act aims to ensure that South Africa's water resources are protected, used, developed, conserved, managed, and controlled sustainably and equitably for the benefit of all people [18]. These authors further show that the Act recognizes that water is a scarce and unevenly distributed national resource and that past discriminatory laws and practices have prevented equal access to and use of water resources. In South Africa, the right to access to sufficient water is guaranteed by Section 27(1)(b) of the 1996 Constitution of the Republic of South Africa. The Constitution requires the state to progressively adopt reasonable legislative measures to realize this right within the limits of its available resources.

The National Water Act establishes the national government's overall responsibility and authority for the nation's water resources and their use. This includes the equitable allocation of water for beneficial use, the redistribution of water, and international water matters. The Act also provides for the integrated management of all aspects of water resources and, where appropriate, delegates management functions to a regional or catchment level to enable everyone to participate [9]. In addition to these legal frameworks, South Africa has implemented various policies and programs to ensure equitable access to water resources. For example, South Africa has a policy called Free Basic Water Access, which entitles every citizen to a certain amount of water regardless of their ability to pay for it. This policy defines the amount of entitlement as 6000 liters per household per month [20].

### 3.4. Community Water Projects in South Africa

Several community water projects in South Africa have been established to provide access to clean water and improve water security for communities. Various organizations, including government agencies, non-profit organizations, and private companies, implement these projects. They often involve the installation of infrastructure such as boreholes, water treatment systems, and pipelines to provide communities with access to clean water. Some projects also focus on education and building capacity to help communities sustainably manage their water resources [20,38].

Some examples of these projects are listed below:

Coca-Cola Beverages South Africa (CCBSA) has implemented a Water Stewardship Strategy 2030 focusing on regenerative operations, healthy watersheds, and resilient communities. The company has implemented several projects to help South African communities become more water resilient, including Coke Ville, which brings solar-powered groundwater harvesting and treatment to communities experiencing water insecurity [19].

Nustad also mentions the Community Water project by Awqaf SA, which aims to provide water to communities in remote parts of South Africa by installing boreholes. This project was made possible via Ahmed Mohamed Baghlaf Waqf sponsorship [21]. The National Water and Sanitation Master Plan (NW & SMP) is a government initiative that

sets out a list of key programs, projects, and actions to be implemented for the protection and development of the national water resources and for the provision of adequate and reliable water and sanitation services for all citizens [22].

*3.5. Water Quality Classification in South Africa*

In South Africa, water quality is regulated by the Department of Water Affairs and Forestry, which has published a series of South African Water Quality Guidelines (as listed in Table 1). These guidelines are used to develop materials to inform water users about the physical, chemical, biological, and aesthetic properties of water. The guidelines consist of water quality criteria, the target water quality range (TWQR), and supporting information, such as the occurrence of the constituent in the environment. The South African National Standard (SANS) 241 Drinking Water Specification states the minimum requirements for potable water to be considered safe for human consumption. Water quality can be classified as Ideal (Class 0) or Good (Class 1) according to the Water Research Commission (WRC) Domestic Use Standard [23].

**Table 1.** DWS water quality "fitness for use" classes currently used in South Africa [23].

| Water Use | Categorization | Description |
|---|---|---|
| Domestic | Class 0 | Water of ideal quality, which has no health or aesthetic effects and is suitable for lifetime use without negative effects. No treatment necessary. |
| | Class 1 | Water of good quality, suitable for lifetime use with few health effects. Some aesthetic effects may be apparent. Home treatment will usually be sufficient. |
| | Class 2 | Water, which poses a definite risk of health effects following long-term or lifetime use. However, health effects are uncommon and unusual following short-term or emergency use. Treatment will be required to render the water fit for continued use. |
| | Class 3 | Water is unsuitable for use, especially by children and the elderly, as health effects are common. Conventional or advanced treatment necessary. |

## 4. Materials and Methods

Utilizing a case study methodology, an in-depth examination and analysis were conducted on two distinct rural sites in South Africa. The primary research question guiding the study was "Can integrated and sustainable water and sanitation systems be effectively implemented in rural community settings?"

Primary documents reviewed included project reports, mid-term evaluation reports, site geological assessments and reports, and school records. Secondary sources utilized were the World Bank Water Data Portal, supported by the Global Water Security & Sanitation Partnership (GWSP) and Google Scholar. The portal is a comprehensive resource that combines water data from various institutions, offering insights on water resources, supply, sanitation, irrigation infrastructure and services, and climate change-related data. Search themes deployed on Google Scholar included local water systems, community participation in water projects, integrated water systems, agroecology and resource utilization, and the water–energy nexus. The inclusion criteria for the literature review encompassed studies on community water and sanitation projects (in South Africa, Africa, and globally), community participation in community-based projects in rural settings, drinking water, water quality and monitoring, infrastructure and resilience, water rights and access, and water ecosystems. Data from the included studies were extracted, including details such as study design, sample size, interventions, outcomes, and results. These data were then synthesized and analyzed using MS Excel by assigning codes to common themes. The results were interpreted in the context of the overall evidence. The findings were reported transparently and completely.

## 5. Results of the Study in Relation to the IWS

### 5.1. Applying Structured Work Packages (WP)

The two projects were implemented using structured work packages (WPs), which are designed to deliver a valuable product or service and ensure its sustainability [23]. The first, Stakeholder Engagement and Facilitation, involves identifying key stakeholders, understanding their interests and influence, and engaging them in the project. Facilitation makes tasks easier for others, often involving problem-solving or decision-making processes. The second WP, Detailed Assessments and Audits, involves a thorough evaluation of various aspects of the project. Audits are conducted to ensure compliance with specified standards or regulations. The Project Management and Coordination WP include planning, organizing, and managing resources to successfully achieve specific project goals and objectives. It involves coordinating tasks, resources, and stakeholders to keep the project on track. The Resource Allocation WP involves assigning and managing assets in a manner that supports the project's goals. Resources include personnel, equipment, financial resources, and other necessary inputs [33]. The Training and Capacity Development WP focuses on enhancing knowledge and skills transfer among the project team and stakeholders. It includes formal training sessions, workshops, coaching, and other learning opportunities. This is followed by applying interventions and specific actions or programs to achieve the project's goals. A key WP is Governance, Power Relationships, and Conflict Resolution. Good governance is essential for decision-making in the project and for determining who the decision-makers are. Power relations in community-based projects encompass the power dynamics within the community, which has a major influence on the project. Conflict resolution addresses diverse interests and perspectives often present within a community. Monitoring, Evaluation, and Learning facilitates tracking the project's progress against its objectives. Evaluating the project's impact or effectiveness is an essential outcome. Lessons learned from the findings enhance future decision-making and practice. Each work package is crucial in ensuring a project's successful implementation and sustainability [34].

### 5.2. Eastern Cape School Support Programme

The Eastern Cape Schools Program (ECSP) project was implemented to enhance the feeding scheme for learners within the water–energy–food nexus. It thus had to also ensure sustainable water and sanitation at the Bijolo, Gwebinkundla, and Mqanduli primary schools (Figure 2 shows the locality of the three schools). These schools are in areas experiencing high water and food shortages. Rural schools face severe challenges, including insufficient funding from the state, dilapidated infrastructure, and lack of resources [34] (Table 2).

**Table 2.** Borehole profiles.

| Borehole ID | Resource Type | Latitude S | Longitude E | Use | Discharge Rate (L/s) | Remark |
|---|---|---|---|---|---|---|
| EC-T20-164 | Borehole | 31.72270 | 28.72270 | Not in use | Low discharge | The discharge rate for the 3 boreholes was very low, and it was recommended to use alternative water systems such as rainwater harvesting. |
| EC-T20-1275 | Borehole | 28.72332 | 28.72332 | Not in use | Low discharge | |
| EC-T20-1277 | Borehole | 28.72280 | 28.72280 | Not in use | Low discharge | |

The ECSP incorporated essential aspects, including stakeholder engagement and partnerships. Stakeholder engagement is critical for strategic planning as it contributes to personal development, co-designing projects, transparent decision-making, managing expectations, reducing project risks, and building trust [25]. Stakeholder mobilization involved meetings with school principals, teachers and the school governing bodies, local councilors, headmen, and chiefs. A formal agreement with the Eastern Cape Department

of Education (ECDoE) established a strong and productive partnership. The ECDoE contributed funding to the project and infrastructure. In addition, there was active involvement and participation from the EC Department of Rural Development and Agrarian Reform. The project also mobilized resources with experience and expertise from the University of Johannesburg and the Walter Sisulu University, which has a campus in Mthatha close to the schools. In keeping with the structured research approach of the project, an initial detailed assessment of water and energy was conducted. JG Afrika, a reputable water company, was contracted to conduct a geohydrological assessment. The company's report (2019) showed that the principal groundwater occurrence was from an intergranular and fractured aquifer type, with median borehole yields ranging between 0.5 and 2.0 liters per second (L/s). The groundwater potential within the project area was considered "poor", as confirmed by a hydro census conducted by Dhliwayo [26]. Three existing boreholes were identified within a 2 km radius of Gwebinkudla JSS. No existing groundwater resources were found at Mqanduli and Bijolo JSS. The discharge rate of the three identified boreholes was classified as low, and it was recommended not to refurbish or dig deeper. Alternative water management systems, such as stormwater retention, roof water harvesting, and stepped platforms, were recommended. Water samples from the three schools were analyzed and showed high levels of Escherichia coli (*E. coli*) since the water at the schools was untreated. Bacterial contamination of drinking water is a major public health problem in rural areas of sub-Saharan Africa. Unimproved water sources are a major cause of severe diarrhea in humans [27]. This study assessed *E. coli* counts in drinking water from different sources and their relationship with water source protection status and neighborhood sanitation and hygiene practices.

One of UJ's Senior Research Associates, an environmental architect, also representing Advance Environmental Design Initiatives (AEDI), which offers adaptive technology, designed a stepped platform and stormwater retention model for the schools. Water treatment and conservation awareness sessions were conducted with teachers and learners. Microbial treatment products were introduced at the three schools to teach the learners the importance of treating the water before drinking it and to encourage them to conserve water and maintain hygienic and clean toilets.

Smart water management systems are one way to achieve sustainable water management. These systems integrate alternative water management systems to improve water supply and availability [27].

The AEDI report (2019) proposed a spatial integration and land management program, including a water management plan. The aim of the plan was to transform the environments of the three schools to improve adaptive capacities and incubate sustainable futures for both school children and community members. Proposed solutions (Figure 4) for the schools included stepped management using stepped platforms, stormwater retention, and roofwater harvesting. The platforms incorporate both bio-filtration and bio-retention systems for land and soil management, preventing erosion and enabling water harvesting for agricultural production. The proposal takes advantage of stormwater flows originating from the street by channeling them into a stormwater retention system. This agricultural water can be managed throughout the year to produce seasonal fresh food for the school kitchen. The new roof structures can also contribute to providing portable drinking water.

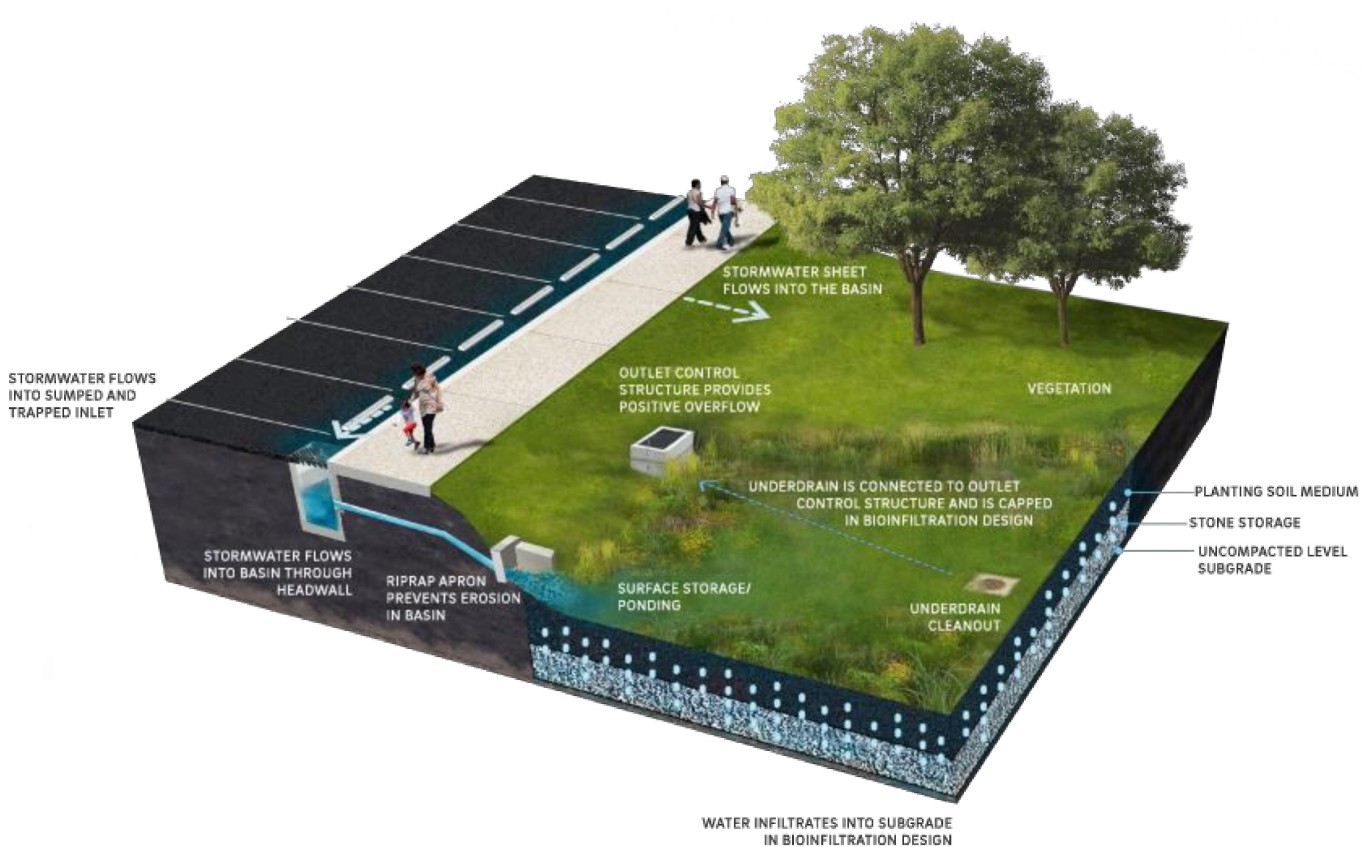

**Figure 4.** Integrated water management system.

### 5.3. Sanitation

Poor sanitation is a major issue in many schools in South Africa, especially in rural areas. Over 6700 education sites have pit latrines, which are considered inadequate sanitation [27]. At Gwebinkudla JSS, children and staff resorted to using nearby bushes as toilets due to the dilapidated state of the existing facilities. The project undertook the renovation of the existing toilets and subsequently collaborated with the ECDoE to install new and upgraded pit toilets, thereby restoring dignity for the school's pupils and staff. Clean and safe toilets not only bring dignity but also hope to learners and poor communities. One sustainable solution recommended is the installation of eco-toilets. These toilets are environmentally friendly, use minimal or no water, and are economically efficient, saving both water and energy resources [30].

### 5.4. Kleenhealth Microbial Products

In recent years, microbial cleaning products have received much attention. Kleenhealth microbial products were applied at the schools and proved to be more environmentally friendly, as they use renewable resources and biodegrade naturally without leaving harmful residues. Moreover, these products are safer for human health as they do not contain toxic or corrosive substances that can irritate the skin, eyes, or respiratory system. They offer versatility and cost-efficiency by serving various cleaning purposes, reducing the need for multiple products. Kleenhealth advocates for innovative and eco-friendly solutions and highly values social equity, economic efficiency, and environmental sustainability. Their products offer long-lasting protection against multiple COVID-19 strains and other harmful bacteria [31].

### 5.5. Phumulani Agri-Village (PAV)

The Phumulani Agri-Village faces acute water scarcity challenges due to recurrent droughts, degradation of surface water resources, and increasing demand for water for

agroecology projects, as well as the 32 households. These challenges mirror those faced across the Mpumalanga province. More than half of Mpumalanga's water supply infrastructure is in a state of disrepair. Millions of liters of fresh drinking water are lost due to leaks and vandalism. Even where water is available, 71% of the supply systems fail to meet the required safety levels [29,32].

Collaborating with the Exxaro company and the Community Property Association (CPA), the project facilitated stakeholder engagement via a workshop. This initiative aimed to strengthen cooperation among water and sanitation stakeholders in the Belfast Local Municipality. The goal was to improve water and energy efficiencies and subsequent provision of water to enhance the economic growth and development for the 32 households in PAV. Additional stakeholders included SGD NPO, Ion Exchange, WellsAfrica, and surrounding farmers. The outcomes of the consultation are outlined below.

- Synergize plans: Use the 4IR and engineering values guided by carbon emission performance targets.
- Allocate water efficiently: Distribute water efficiently and effectively to maximize outputs, outcomes, and throughput.
- Implement energy efficiency strategies: Install energy efficiency measures, including appropriate waterworks and CCTV cameras.
- Promote water reuse: Train the 32 households on water reuse techniques to minimize waste.
- Maximize water productivity: Enhance water productivity via strategic planning and implementation.
- Identify gaps: Identify gaps related to water and energy requirements.
- Promote Phumulani Agri-Village concept: Promote the Phumulani Agri-Village concept to enhance green smart villages within the framework of agroecology.

### 5.6. Water Management at PAV

A hydro census was conducted across the PAV area in August 2022. The survey focused exclusively on the PAV property, aiming to identify existing boreholes to enhance our knowledge and understanding of borehole locations, groundwater systems, and current groundwater usage. See Figure 5 for the location of the boreholes.

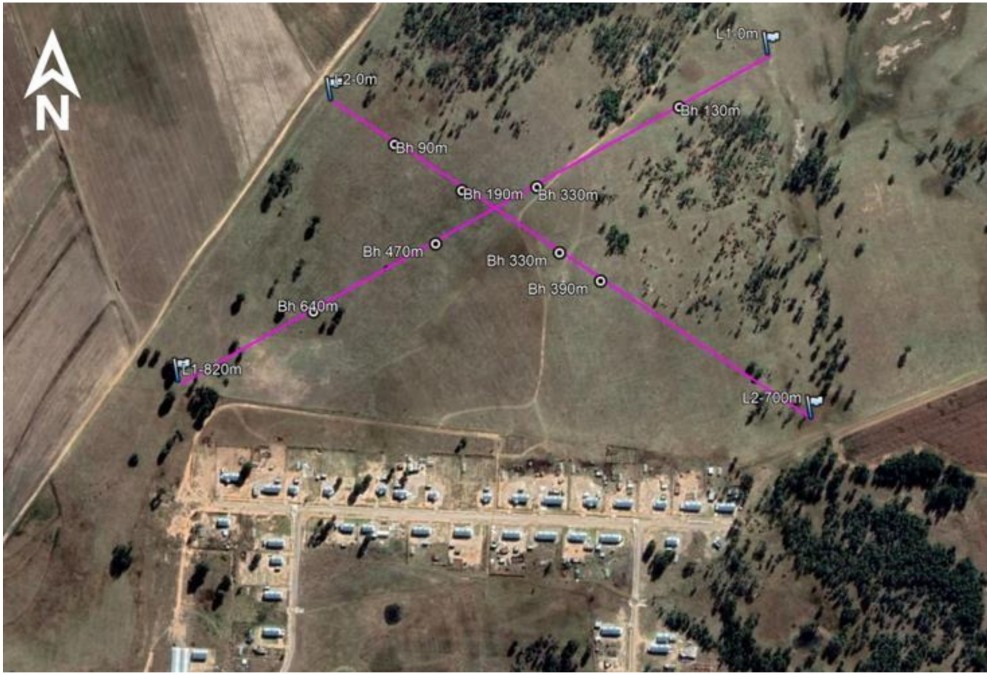

**Figure 5.** Survey Layout and locality map for the boreholes.

The collected water samples were analyzed for basic inorganic parameters and bacterial counts. The results were then compared against the drinking Water Standards [39]. Table 3 presents the water quality results and provides a summary of the available volume for all boreholes at PAV. Boreholes PUM2, PUM3, PUM6, PUM7, and BBBH1 exhibited very low yields. It is important to note that the yield calculations for these boreholes are based on data obtained from a single aquifer test.

**Table 3.** Borehole profiles at PAV.

| Borehole ID | Static WL (mbgl) | Borehole Depth (m) | Recommended Pump Installation Depth (mbgl) | Borehole Yield (L/hour) | Recommended Abstraction Rate (Limited to 4 to 8 h of Pumping per Day) |
|---|---|---|---|---|---|
| PUM1 | | | Not tested | | |
| PUM2 | 6.80 | 78 | -- | 100 | Not recommended for use |
| PUM3 | 7.63 | 13 | -- | unknown | Not recommended for use |
| PUM4 | 19.30 | 82 | 75 | 500 | 500 L/hour |
| PUM5 | 18.60 | 82 | 75 | 650 | 650 L/hour |
| PUM6 | 2.51 | 11 | -- | unknown | Not recommended for use |
| PUM7 | 2.20 | 32 | -- | 100 | Not recommended for use |
| PUM8 | 19.50 | 77 | 70 | 500 | 500 L/hour |
| PUM9 | | | filled with rocks | | |
| PUM10 | | | filled with rocks | | |
| BBBH1 | 10.17 | 80 | 75 | 200 | 200 L/hour |
| GWA-BH1 | 32.57 | 138.7 | 100 | -- | 720 L/hour |
| GWA-BH2 | 20.73 | 125.6 | 105 | -- | 396 L/hour |
| Cumulative yield: | | | | | 2966 L/hour |
| Approx. 24,000 L per day, if only pumped for 8 h per day, allowing for borehole recovery. | | | | | |

*5.7. Water Quality and Management at PAV*

Good water quality benefits the environment and public health. The lack of safe and clean water access is one of the major risk factors for spreading infectious diseases, as highlighted by the recent cholera outbreak in Hammanskraal, which resulted in numerous deaths [1,23].

Based on the SANS241 drinking water guideline and the sampled borehole water results, groundwater from the two community water supply boreholes (PUM4 and PUM5) is NOT suitable for human consumption, and treatment is recommended. This is mainly due to the elevated *E. coli* count in both boreholes. The water quality could improve with frequent use, especially in terms of reducing the turbidity count. To address this, a water treatment plant has been proposed to improve the quality of the water.

Ion Exchange Safic (IES), an internationally renowned water company, conducted a site visit and survey of PAV and recommended a schematic process flow diagram (PFD). This diagram ensures that the water supply from the boreholes is piped to a feed water tank (TK01) to service the water treatment plant, which will be located adjacent to the steel storage tank (TK02) on site. The water from the feed tank will be supplied via a feed pump for treatment in the containerized water treatment plant. The treated water can then be supplied separately from the containerized plant into the steel storage tank for community supply and the Agriproject. Figure 6 illustrates this proposed setup.

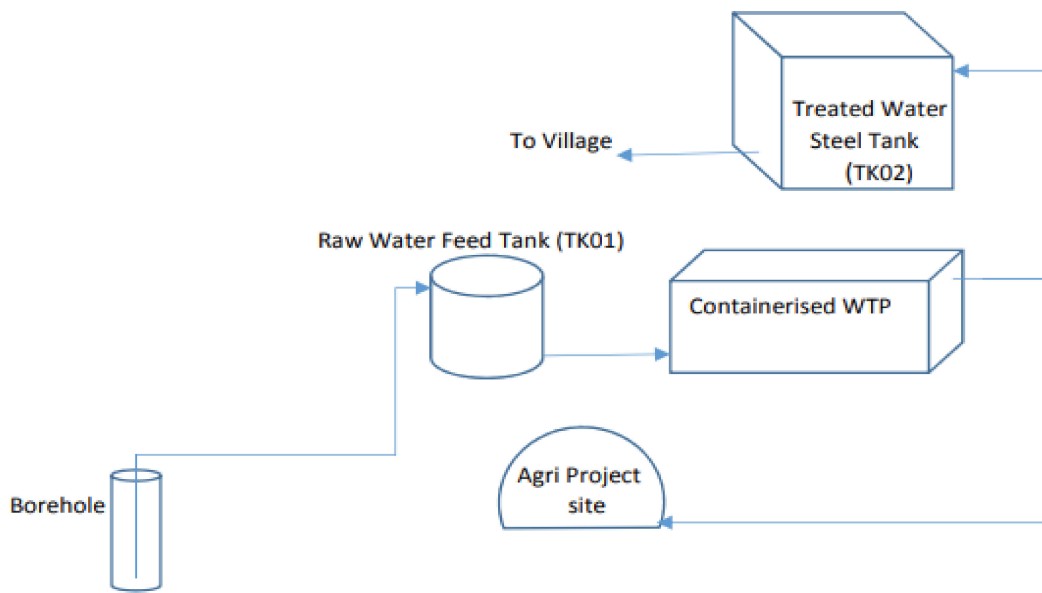

**Figure 6.** Proposed water plant infrastructure at PAV.

## 6. Discussion of the Results

Implementing structured work packages (WPs) at the Eastern Cape Schools and Phumulani Agri-Village clearly demonstrated the importance of effective program execution, particularly with regard to water and sanitation aspects. These assessments and audits of WP provided a comprehensive understanding of potential risks and vulnerabilities, which improved compliance with various entities such as municipalities, the Department of Education, and the Department of Agriculture and Rural Development. The Stakeholder Engagement and Facilitation WP contributed significantly to building good relationships with the communities and partners, resulting in co-designing projects, transparent decision-making, managing expectations, reducing project risks, and building trust. The Project Management and Co-ordination WP ensured successful project implementation and efficient resource utilization. The Training and Capacity Development WP enhanced the knowledge and skills of project beneficiaries and local communities regarding water hygiene, maintaining clean and hygienic toilets, and promoting hand washing. These positive achievements were attained despite various challenges related to governance, limited funding, and conflict resolution. WP seven contributed to improved internal and external relationships via the establishment of new structures and processes. The Monitoring and Evaluation WP facilitated tracking progress, continuous learning, and reporting on project outcomes and impact.

Despite the initial benefits, Phumulani Agri-Village continues to grapple with severe water scarcity issues due to droughts, degradation of surface water resources, and increasing water demand for agroecology projects and the 32 households. These challenges are reflective of the broader water issues in the Mpumalanga province. To address these issues, a workshop was organized to strengthen cooperation among various stakeholders, including the Exxaro company, the Community Property Association, Siyakhana NPO, IonExchange and WellsAfrica (service providers), local government representatives, and surrounding farmers. The workshop facilitated a better understanding of the water issues and led to the development of short-, medium-, and long-term action plans.

This consultation involved plans to utilize 4IR and engineering solutions guided by carbon emission performance targets, enhancing water efficiency, identifying gaps in water and energy requirements, and promoting a green, smart ecological village.

A water treatment plant was proposed by IonExchange Safic, an internationally renowned water company, following a site visit and survey. This water treatment plant will be implemented soon and will improve the efficiency and reliability of the water supply at PAV.

Water treatment and conservation awareness sessions were conducted with teachers and learners at the schools. Microbial treatment products were introduced to teach

learners the importance of keeping pit toilets clean and hygienic. A stepped platform and stormwater retention model for the schools were designed. Implementing smart water management systems is one way to achieve sustainable water management. A proposed sustainable solution is the installation of eco-toilets, which are environmentally friendly and use minimal water. The use of microbial cleaning products is recommended. These products use renewable resources, biodegrade naturally without leaving harmful residues, and are safe for human health.

This paper compares its findings with another case study in Limpopo titled "Barriers to Water and Sanitation Safety Plans in Rural Areas of South Africa" [31]. The case study examines the implementation of Water and Sanitation Safety Plans and suggests that understanding the factors hindering implementation could significantly improve water and sanitation projects. As discussed in our study, implementing work packages has proven to be an effective tool for successfully carrying out water and sanitation projects. Additionally, in the Limpopo study, the lack of access to purified water and inadequate sanitation facilities have led to reported waterborne diseases within the communities. The case study suggests that educating communities and upgrading or building new modern infrastructure is essential. Our study at the two sites in this paper aligns with infrastructural development and knowledge and skills transfer. However, our research also examines power relations and governance structures, monitoring, and evaluation. These additional aspects provide a more comprehensive understanding of the factors influencing the success of water and sanitation projects. According to a study by Montwedi et al. (2021), sanitation and water shortage challenges require a paradigm shift. This involves the transition of solid waste into biogas plants and the use of multiple water sources, such as rainwater harvesting and boreholes, all of which are being implemented. PAV has installed a biogas plant as part of its circular economy approach [36].

### 7. Conclusions and Recommendations

The findings from the three Eastern Cape schools highlighted the poor groundwater potential in the project area. To address this problem, rainwater harvesting systems were installed at the schools via new gutters and water tanks at each school. Additionally, newly upgraded pit toilets were installed at one school, and effective microbial products were applied to improve sanitation at all three schools. Furthermore, stepped platforms and stormwater retention models were designed and are likely to be implemented soon. The collected stormwater can be used for agricultural purposes and managed throughout the year to create seasonal fresh food produce for the school kitchen. In the context of Phumulani Agri-Village (PAV), a comprehensive approach was adopted. By combining the cumulative yield of several boreholes, an adequate water supply will be provided at PAV. Easier access and water quality are being addressed by installing solar pumps and a containerized water treatment plant. The treated water can then be supplied directly to the PAV households and the agricultural project. The installation of new eco-toilets is highly recommended to further improve hygiene, safety, and water conservation. These interventions have positively impacted the living conditions of the two communities. By providing better water and sanitation systems, they have not only alleviated immediate concerns but also restored dignity among the vulnerable population. Looking ahead, the installation of the biogas plant will demonstrate commitment to further social, economic, and environmental sustainability, ensuring a holistic approach to community development [38].

**Author Contributions:** Conceptualization, M.J.R. and E.M.; Writing—original draft, M.J.R.; Writing—review & editing, M.J.R. All authors have read and agreed to the published version of the manuscript.

**Funding:** This research received no external funding.

**Data Availability Statement:** The data presented in this study are available on request from the corresponding author.

**Conflicts of Interest:** The authors declare no conflict of interest.

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
