# Peer review of "Integrated and Sustainable Water and Sanitation Systems at Two Rural Sites in South Africa"

_water, doi:10.3390/w15223974_

Round 1

Reviewer 1 Report (Previous Reviewer 3)

Comments and Suggestions for Authors

The paper entitled "Integrated and sustainable water and sanitation systems at two rural sites in South Africa"  proposes  a novel study about water sustainability. The proposition is important. However, authors must do changes according to the following notes and remarks:

- The section 2- Objectives can be included in the Itroduction ; as it contains only two short paragraphs

- I suggest also to add a short paragraph at the end of the introduction that give the structure of the rest of the paper.

- Why you don't discuss the obtained results.

- The comparison of your proposed work with other  related ones  is necessary

- Why the conclusion contains many short chapters

Author Response

he section 2- Objectives can be included in the Itroduction ; as it contains only two short paragraphs

Comment:  I suggest also to add a short paragraph at the end of the introduction that give the structure of the rest of the paper.

Response: We have added the short paragraph giving the structure of the paper

Comment: Why you don't discuss the obtained results.

Response: we have added discussion of results section

Comment: The comparison of your proposed work with other  related ones  is necessary

Response: We have added a paragraph discussing and comparing similar work a

Comment: Why the conclusion contains many short chapters

Response: We have consolidated the conclusion section

Reviewer 2 Report (Previous Reviewer 4)

Comments and Suggestions for Authors

1. The beginning of the abstract should mention the research background of this paper.

2. There may be some problems with the allocation of paragraphs in the introduction; in addition, the introduction should have to state the innovative nature of the study and the scientific problems that need to be addressed in this study.

3. It is recommended that a regional overview map be added to Part III.

4. Please check the labeling of the subsections of the manuscript, e.g. line 145.

5. In the literature review, only the South African water quality classification, legislation situation, and community water programs are described. it is recommended that the literature review take a global perspective.

6. The manuscript needs to add a discussion section, in addition, the structure of the article needs to be more confusing and it is recommended that it be redistributed.

7. What are the limitations of the study?

Comments on the Quality of English Language

Moderate editing of English language required.

Author Response

Comment 1: The beginning of the abstract should mention the research background of this paper.

Response: we have added the research background

Comment 2:  There may be some problems with the allocation of paragraphs in the introduction; in addition, the introduction should have to state the innovative nature of the study and the scientific problems that need to be addressed in this study.

Response: We have readjusted the paragraphs in the introduction. Problem statement and the nature of the study was well articulated. 

Comment 3: It is recommended that a regional overview map be added to Part III.

Response: regional overview map was added to Part III. 

Comment 4. Please check the labeling of the subsections of the manuscript, e.g. line 145.

Response: labelling of subsection was redone

Comment 5. In the literature review, only the South African water quality classification, legislation situation, and community water programs are described. it is recommended that the literature review take a global perspective.

Response: literature on global perspective was added. 

Comment 6. The manuscript needs to add a discussion section, in addition, the structure of the article needs to be more confusing and it is recommended that it be redistributed.

Response: We have reworked the structure of the paper and also added the discussion section. 

Comment 7. What are the limitations of the study?

Response: We added a sentence on the limitation of the study. 

Reviewer 3 Report (Previous Reviewer 2)

Comments and Suggestions for Authors

The paper discuss the effectiveness of water and sanitation programs in two community settings in South Africa. The findings aims at supporting water and sanitation interventions by applying the integrated Water Management Systems.

The topic is relevant and the results of the case study are significant. The paper has been improved respect the previous submission.

However, in order to provide scientific soundness, the paper needs to be integrated as follows:

1)      The introduction do not clearly present the contents and the aims of the paper. It should provide the background and context of the study and relate it to the purpose of the work, in order to highlight its significance. The state of the art should be introduced with references to the main publications in the field. In the paragraph generally is briefly mentioned the structure of the paper, the main aims and the expected results, in order to guide the reader through the research process.  

2)      Objectives should be included in the introduction and related to the paper structure. The research question (pag 6 line 242) needs to be anticipated in the introduction.

3)      The literature review should be placed before the description of the study site. It should also include references regarding the topic in general, the scientific debate, the methodological approach.

4)      Apparently the research methodology includes a systematic literature review and empirical study through two case studies. Is mandatory to clarify this process. In paragraph 4 there is a sectorial literature review, while in paragraph 5 a generic description of a Systematic Literature Review is introduced in a few lines. Please clarify if a SLR has been done, the keywords used for the research and the results. Lines 245-261 explain the activities caried out without presenting any result. In addition, there are only 35 references in the paper. The SLR should include an annex with all the literature collected.

5)      It would be appropriate to explain the criteria for the selection of the case study while describing the study site.

6)      The results need to be thoroughly organized, in order to better understand the findings of the empirical analysis, the findings of the participatory process and the relationships with the literature results. I could be useful add a paragraph dedicated to the discussion.

7)      Considering the relevance of the topic, the conclusions should include a possible trajectory for  generalization and policy implications.

I suggest to rethink the titles of the paragraphs in order to better express the contents.

Comments on the Quality of English Language

The English is generally adequate. 

However, there are some mispelling and a few sentences that need to be clarified.

Author Response

Comment 1: The introduction do not clearly present the contents and the aims of the paper. It should provide the background and context of the study and relate it to the purpose of the work, in order to highlight its significance. The state of the art should be introduced with references to the main publications in the field. In the paragraph generally is briefly mentioned the structure of the paper, the main aims and the expected results, in order to guide the reader through the research process.  

Response: We have reworked the introduction highlighting the purpose of the study and its significance added references and also redone the structure of the paper. 

Comment 2)      Objectives should be included in the introduction and related to the paper structure. The research question (pag 6 line 242) needs to be anticipated in the introduction.

Response: We have included the objectives in the introduction as well as the research question. 

Comment 3)      The literature review should be placed before the description of the study site. It should also include references regarding the topic in general, the scientific debate, the methodological approach.

Response: Comments from the first reviewer asked us to move it to the current position

Comment 4)      Apparently the research methodology includes a systematic literature review and empirical study through two case studies. Is mandatory to clarify this process. In paragraph 4 there is a sectorial literature review, while in paragraph 5 a generic description of a Systematic Literature Review is introduced in a few lines. Please clarify if a SLR has been done, the keywords used for the research and the results. Lines 245-261 explain the activities caried out without presenting any result. In addition, there are only 35 references in the paper. The SLR should include an annex with all the literature collected.

Response: we feel we have adequately addressed the issue of the methodology. 

Comment 5)      It would be appropriate to explain the criteria for the selection of the case study while describing the study site.

Response: We have included and explained the criteria for the selection of the two sites 

Comment 6)      The results need to be thoroughly organized, in order to better understand the findings of the empirical analysis, the findings of the participatory process and the relationships with the literature results. I could be useful add a paragraph dedicated to the discussion.

Response: we have included the discussion of the results section which further explains the results. 

Comment 7)      Considering the relevance of the topic, the conclusions should include a possible trajectory for  generalization and policy implications.

Response: we have included policy implications in our conclusion.

Comment I suggest to rethink the titles of the paragraphs in order to better express the contents.

Response: we have reworked the titles of the paragraphs.

Reviewer 4 Report (Previous Reviewer 1)

Comments and Suggestions for Authors

No comments. No changes in red

Comments on the Quality of English Language

Minor changes

Author Response

Thank you for the wonderful review. Please see the updated paper. 

Round 2

Reviewer 1 Report (Previous Reviewer 3)

Comments and Suggestions for Authors

Author has done requested changes

Author Response

Thank you for reviewing our paper. Please see the updated draft. We have made some minor changes to the methodology and the references. 

Reviewer 3 Report (Previous Reviewer 2)

Comments and Suggestions for Authors

The paper has been improved. The case studies are interesting.

Nevertheless, in order to be feasible for publication, the methodology needs to be better addressed. 

The declared SLR "In the Materials and Methods section, the Systematic Literature Review (SLR) is conducted using the PRISMA protocol (line 140)" does not fit with the recognized protocol of this approach. The paper definitely does not include a SLR. The literature review itself is still limited, but fit with a case-study approach.

According to this, I suggest to focus on the fieldwork, by enhancing the scoping lit rev on the topic and on the context and describing more in depth the cases. I suggest to remove the references to a missing SLR as well.

Comments on the Quality of English Language

Adequate English

Author Response

We have addressed the SLR issue and emphasised the case study approach. We also removed the SLR reference and did some further minor editing. 

This manuscript is a resubmission of an earlier submission. The following is a list of the peer review reports and author responses from that submission.

Round 1

Reviewer 1 Report

Comments and Suggestions for Authors

This paper is very interesting, but need more work: (1) need a map of situation of research areas; (2) need a a descriptive situation of environmental micro situation and problems in selected areas; (3) in the introduction please quoted the Un program 2030 and the significance in South Africa, and more international paper on others countries in Africa and in the Global South; (4) please introduce and use the concept of resistance and power relations in selected research areas and in the conclusion, and in finally (5) please enlarged the objectives with use of literature.

Comments on the Quality of English Language

MInor

Author Response

We have reworked the paper situational analysis of the research areas have been includes this include the description of the environmental micro situation and problems in selected. We have added more information on the significance of the SGD goals in South Africa, Africa and globally. The concept of resistance and power relations has been included as part of our work packages. The objectives were expanded.

Reviewer 2 Report

Comments and Suggestions for Authors

The paper deals with the analysis of effectiveness of water and sanitation programs in two community settings in South Africa. The topic is relevant and well presented. However, the contents needs to be integrated and reworked. I suggest:

The declared  transdisciplinary literature review developed with the SLR PRISMA protocol needs to be explained, respecting the procedure for SLR articles.

Consider better explaining the participatory process with the village community.

Conclusions should include generalization purposes and follow up.

Comments on the Quality of English Language

minor misstypings and text reorganization

Author Response

We have explained the SLR PRISMA protocol in detail. We also have expanded and explained the participatory process within the village using the work packages. The conclusion has been expanded. We have also done editing of our grammar and sentence construction. 

Conclusions should include generalization purposes and follow up.

Reviewer 3 Report

Comments and Suggestions for Authors

The present paper studies an integrated and sustainable both water and sanitation methods in the south africa rural regions. The problem studied is very important and significant. we apriciat the authors efforts. However, we have some remarks:

1-Author must introduce the problem related to the use of water in world not only in studied regions

2- The section two must be a part of introduction.

3- Why You put the section litterture review after Materials and Methods section

4- Why section Materials and methods contains only 1 paragraph

5 What is the importance of 3.Conceptual Framework section

6- Authors should compare the obtained out come with the results of other similare studies

Author Response

The present paper studies an integrated and sustainable both water and sanitation methods in the south africa rural regions. The problem studied is very important and significant. we apriciat the authors efforts. However, we have some remarks:

We have added more literature on our problem statement including other regions in Africa. We have combined section 2 with the introduction. We have put the material and methods after the literature review. We have expanded the material and methods section giving more detail. 

We have combined the conceptual framework with the literature review section. 

We have included a similar study. 

Reviewer 4 Report

Comments and Suggestions for Authors

I apologize for not fully grasping the core message intended by the manuscript. The manuscript appears more akin to a project investigation report rather than a typical article, and its logical flow is rather chaotic. For instance, it combines elements of a project report with a literature review sourced from Google Scholar. In this context, it is essential to clarify the titles and sources of the project reports and indicate where the source files can be located. Furthermore, concerning the literature review, the parameters of the PRISMA guidelines are not specified, rendering replication unfeasible. Additionally, there should be distinct analyses for each of these aspects, as well as an overall synthesis highlighting their similarities and differences. Unfortunately, these aspects are not adequately addressed in the manuscript, and certain issues should not have arisen in the first place:

Lines 4-5: The authors' affiliations are identical.

Line 15: What does 'PVA' stand for in full?

Line 23: What does '[200]' signify?

Having only one paragraph in the introduction is inadequate; the significance of water and the regional context might be better separated.

The content of the second chapter, "2. Problem Statement," could be merged with the introduction.

There seem to be two instances of the second chapter; please review the overall article structure.

Comments on the Quality of English Language

 Extensive editing of English language required

Author Response

The authors affiliation are identical because we are from the same centre

PAV is Phumalani Agri-Village the second research site. 

We have deleted the 200 was the word count for the abstract.

We have merged the introduction with the problem statement. 

We have reviewed the overall structure. 
